# IL-33 facilitates rapid expulsion of the parasitic nematode *Strongyloides ratti* from the intestine via ILC2- and IL-9-driven mast cell activation

Jana Meiners[1][☉], Martina Reitz[1][☉], Nikolas Rüdiger[1][☉], Jan-Eric Turner[2], Lennart Heepmann[1], Lena Rudolf[1], Wiebke Hartmann[1], Henry J. McSorley[3], Minka Breloer[1,4]*

1 Bernhard Nocht Institute for Tropical Medicine, Section of Molecular Biology and Immunology, Hamburg, Germany, 2 III. Department of Medicine and Hamburg Center for Translational Immunology, University Medical Center Hamburg-Eppendorf, Hamburg, Germany, 3 Division of Cell signaling and Immunology, School of Life Sciences, University of Dundee, Dundee, United Kingdom, 4 Department of Biology, University of Hamburg, Hamburg, Germany

☉ These authors contributed equally to this work.
* breloer@bnitm.de

**Data Availability Statement:** All relevant data are within the manuscript and its Supporting Information files.

## Abstract

Parasitic helminths are sensed by the immune system via tissue-derived alarmins that promote the initiation of the appropriate type 2 immune responses. Here we establish the nuclear alarmin cytokine IL-33 as a non-redundant trigger of specifically IL-9-driven and mast cell-mediated immunity to the intestinal parasite *Strongyloides ratti*. Blockade of endogenous IL-33 using a helminth-derived IL-33 inhibitor elevated intestinal parasite burdens in the context of reduced mast cell activation while stabilization of endogenous IL-33 or application of recombinant IL-33 reciprocally reduced intestinal parasite burdens and increased mast cell activation. Using gene-deficient mice, we show that application of IL-33 triggered rapid mast cell-mediated expulsion of parasites directly in the intestine, independent of the adaptive immune system, basophils, eosinophils or Gr-1[+] cells but dependent on functional IL-9 receptor and innate lymphoid cells (ILC). Thereby we connect the described axis of IL-33-mediated ILC2 expansion to the rapid initiation of IL-9-mediated and mast cell-driven intestinal anti-helminth immunity.

## Author summary

Parasitic worms leave a trail of destruction while migrating through their host's tissue. Thereby they trigger the release of tissue-derived alarmin cytokines such as IL-33 that promote the initiation of efficient anti-helminth immune responses. Here we use mice infected with the parasitic nematode *Strongyloides ratti* to unravel the chain of events leading from parasite sensing to parasite expulsion. *S. ratti* penetrates the skin of its mammalian host, migrates via skin and muscle tissue to the mouth, is swallowed and reproduces

**Funding:** NR was supported by the Leibniz Center Infection Graduate School (https://www.lc-infection.de/en/).JET is supported by the Priority Programm 1937 "Innate Lymphoid Cells" and the SFB 1192 TP06 both from the German Research Foundation(https://www.dfg.de/en/index.jsp). HJM is supported by a UK Medical Research Council, New Investigator Research Grant MR/S000593/1. (https://mrc.ukri.org/). The funders had no role in study design, data collection and analysis, decision to publish, or preparation of the manuscript.

**Competing interests:** The authors have declared that no competing interests exist.

in the small intestine. The parasite is eventually expelled from the intestine by the action of mast cells that are activated via IL-9. Using inhibitors and enhancers for IL-33 we demonstrate that the release of IL-33 during *S. ratti* infection activates mast cells. Blockade of IL-33 elevated intestinal parasite burden and suppressed mast cell degranulation while stabilization of endogenous IL-33 or application of recombinant IL-33 reduced intestinal parasite burdens and increased mast cell degranulation. IL-33 mediated parasite expulsion independently of adaptive immunity, basophils or granulocytes but dependent on IL-9, innate lymphoid cells and mast cells. In summary we provide an example of how efficient sensing of a tissue-migrating parasite generates a hostile environment in the intestine that facilitates parasite expulsion.

## Introduction

Helminths are large multicellular pathogens that affect one quarter of the human population [1]. They are controlled and eradicated in the context of a canonical type 2 immune response [2]. Thereby, the mammalian immune system needs to initiate rapid and efficient anti-helminth defences while simultaneously organising the repair of inflicted mechanical damage and re-establishing homeostasis to prevent immunopathology.

We use *Strongyloides ratti* [3], a rodent-specific parasitic nematode, to analyze immune responses against a "moving target" which displays tissue migrating and intestinal life stages in the mouse system. Infective third stage larvae (L3) penetrate the skin of their hosts and over the following 2 days migrate via skin, head and lung to the mouth. They are swallowed, reach the intestine and moult to adults by day 5. Immune competent mice terminate the infection in the context of a type 2 immune response within 3–4 weeks and remain semi-resistant to subsequent infections [4]. Eradication of migrating *Strongyloides* L3 during the first 2 days of infection is mediated predominantly by eosinophilic and neutrophilic granulocytes [5–9], whereas efficient expulsion from the intestine is executed by basophils [10] and mast cells [11,12]. Specifically mucosal mast cells are indispensable for the final eradication of *Strongyloides* from the intestine as selectively mast cell-deficient mice are unable to terminate infections for more than 20 weeks [12]. The mast cells are activated via classical antibody- (IgE and IgG) mediated mechanisms once adaptive immunity is established [13]. However, the early mast cell degranulation that facilitates intestinal parasite control during the first week of infection is independent of adaptive immunity and promoted by IL-9 [12,14]. *S. ratti* actively delays this rapid IL-9-dependent, mast cell-mediated expulsion from the intestine via expansion of Foxp3$^+$ regulatory T cells (Treg) [15,16] and induction of negative regulatory receptors such as B and T Lymphocyte Attenuator (BTLA) on T effector cells [17]. Treg and BTLA$^+$ T cells expand during infection and either Treg depletion or absence of BTLA or its ligand, Herpes Virus Entry Mediator (HVEM), results in elevated IL-9 production, accelerated mast cell activation and rapid expulsion of *S. ratti* from the intestine. The factors regulating initiation of this anti-helminth response, however, are still not fully understood.

IL-33 is a tissue-derived nuclear cytokine [18], that was shown to promote type 2 immune responses during allergy and helminth infection [19,20]. Exposure to migrating parasitic nematodes or intranasal application of chitin, the dominant material of the parasites cuticula, triggered transcription and/or release of IL-33 in the lung [21,22] and intestine [23]. A central function for IL-33 in host defence was established by several studies demonstrating that mice lacking either IL-33 or the IL-33 receptor ST2 displayed increased intestinal parasite burdens in *Strongyloides venezuelensis* [22], *Nippostrongylus brasiliensis* [24], and *Heligmosomoides*

*polygyrus* [25] infection, increased muscle larval burdens during *Trichinella spiralis* [26] infection and elevated microfilaremia during *Litomosoides sigmodontis* [27] infection. Further evidence for the importance of IL-33 in anti-helminth immunity arises from the fact that the chronic intestinal parasitic nematode *H. polygyrus* [28] antagonizes IL-33 function via secretion of IL-33 inhibitors [29,30].

Here we employ the *H. polygyrus*-derived IL-33-suppressive alarmin release inhibitor (HpARI) [30] and an IL-33-stabilizing truncation variant of HpARI, CCP1/2 [31], to analyse the role of IL-33 during the early immune response to *S. ratti*. Neutralization of endogenous IL-33 increases intestinal parasite burdens while stabilization of endogenous IL-33 or application of recombinant (rec.) IL-33 reciprocally reduces parasite burdens. The IL-33-triggered reduction of intestinal parasite burdens is not established during tissue migration and is independent of the effector cells controlling the tissue migrating larvae such as eosinophils or neutrophils. We show that IL-33 induces rapid activation of mucosal mast cells, preventing embedment of arriving parasites in the intestine. Accelerated mast cell activation is independent of the adaptive immune system, eosinophils, basophils or neutrophils, but strictly depends on the presence of ILC and a functional IL-9 receptor. In summary, we provide evidence for a non-redundant IL-33-triggered ILC2-, IL-9-, and mast cell-dependent innate pathway facilitating the defense against intestinal helminths during the first week of infection.

## Results

### IL-33 promotes intestinal immunity to *S. ratti*

The parasitic nematode *S. venezuelensis* triggers IL-33 release in the lungs of infected mice [22]. Likewise, lung explants from *S. ratti*-infected mice show significantly increased release of this alarmin cytokine, while small intestinal explants showed a trend for elevated IL-33 release, although this did not reach statistical significance (S1 Fig).

To evaluate the impact of endogenous IL-33 release in initiating the protective immune response to *S. ratti in vivo*, we used the *H. polygyrus*-derived IL-33 inhibitor HpARI (Fig 1). HpARI antagonizes IL-33 function *in vivo* by tethering IL-33 to the nuclear DNA of necrotic cells and by blocking productive IL-33 interaction with its receptor ST2 [30]. Intranasal (i.n.) application of HpARI prior to *S. ratti* infection increased the intestinal parasite burdens while i.n. application of rec. IL-33 reciprocally decreased parasite burdens (Fig 1B). Recently, a truncation of the HpARI full length protein, CCP1/2, was shown to enhance IL-33-dependent responses to allergen administration or *N. brasiliensis* infection through stabilisation of the active cytokine [31], thereby amplifying endogenous IL-33 responses *in vivo*. Application of CCP1/2 prior to *S. ratti* infection reduced intestinal parasite burdens, thus phenocopying application of rec. IL-33 (Fig 1C) and demonstrating the importance of the endogenous IL-33 response during the early stages of infection.

To define the site of IL-33-mediated immunity, we performed a kinetic analysis of intestinal parasite burdens comparing mice that received rec. IL-33 to untreated mice (Fig 2A). After subcutaneous injection into the hind foot pad, the first L3 arrived in the small intestine by day 2 p.i. and maximal numbers were present at day 3 p.i. (Fig 2B). While the numbers of newly arrived intestinal L3 were not affected by IL-33 treatment, viable parasite numbers that were attached to the intestine rapidly declined in the intestine of IL-33-treated mice after day 3 p.i. The presence of *S. ratti*-derived DNA in the faeces of these mice on days 3 and 4 p.i. reflects the *S. ratti* larvae that arrived by day 3 but did not embed successfully in the intestine of IL-33-treated mice (Fig 2C). Untreated mice, by contrast, displayed constant numbers of intestinal parasites after day 3 p.i. that moulted to adults by day 4/5 p.i., demonstrating successful embedding of *S. ratti* (Fig 2B, open circles). In support of this notion, almost no *S. ratti*-

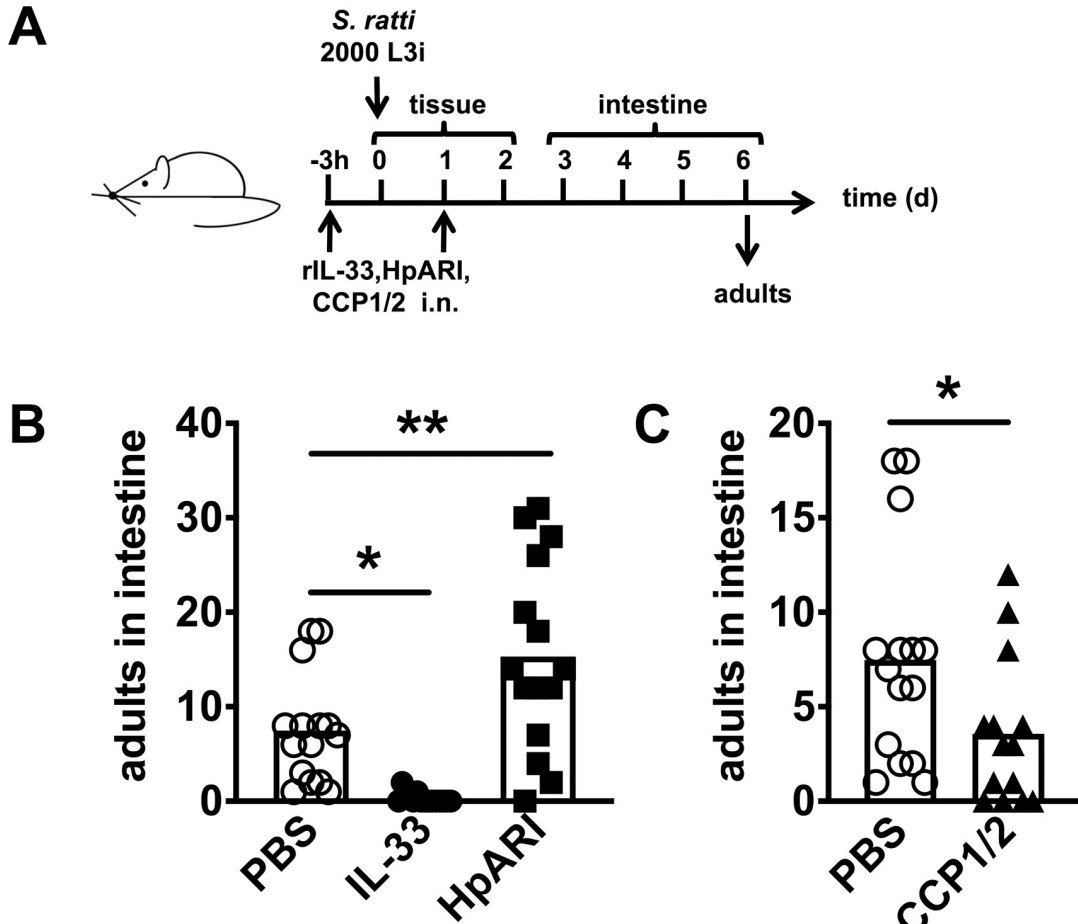

**Fig 1. IL-33 reduces intestinal *S. ratti* burden. (A)** Experimental procedure: BALB/c mice were treated i.n. with PBS (open circles), 1 μg rec. IL-33 (closed circles), 5 μg rec. HpARI (closed squares) or 5 μg rec. CCP1/2 (closed triangles) in 20 μl PBS 3 h before and 24 h post s.c. infection with 2000 *S. ratti* L3. **(B and C)** Shown are combined results from 3 independent experiments (n = 4–5 per experiment and group), each symbol represents an individual mouse, bars show the mean and asterisk indicate statistically significant differences of the means of untreated (PBS) or treated mice (**B**: one-way ANOVA, **C**: students t-test).

derived DNA was detected in the faeces of these mice until day 4 (Fig 2C, open circles). We did not record later time points since *S. ratti* adults start to reproduce by day 5 p.i. and faecal *S. ratti*-derived DNA after day 5 p.i. would predominantly reflect released eggs and L1.

To further distinguish between the impact of IL-33 on immunity during tissue migration and intestinal life stages, we next applied IL-33 at days 4 and 5 of infection, a time point when all *S. ratti* L3 had reached the intestine and developed to L4 or adults (Fig 2A black and blue). Surprisingly, i.n. application of IL-33 after completion of the tissue migration still resulted in the same reduction of day 6 intestinal parasite burdens as induced by i.n. application of IL-33 before *S. ratti* infection (Fig 2D and 2E). In light of this finding we measured if i.n. application of IL-33 would also lead to systemic distribution of the cytokine. We recorded systemic elevation of IL-33 concentration in the serum 3 hours after i.n. and after systemic, intraperitoneal (i.p.) application of rec. IL-33 in non-infected mice (S2A and S2B Fig), suggesting that i.n. application of IL-33 did not selectively target the lung. Indeed, repetition of the *S. ratti* infection experiments using i.p. application of IL-33 phenocopied the i.n. application of IL-33. Parasite burdens in the intestine day 6 p.i. were reduced when IL-33 was applied before infection

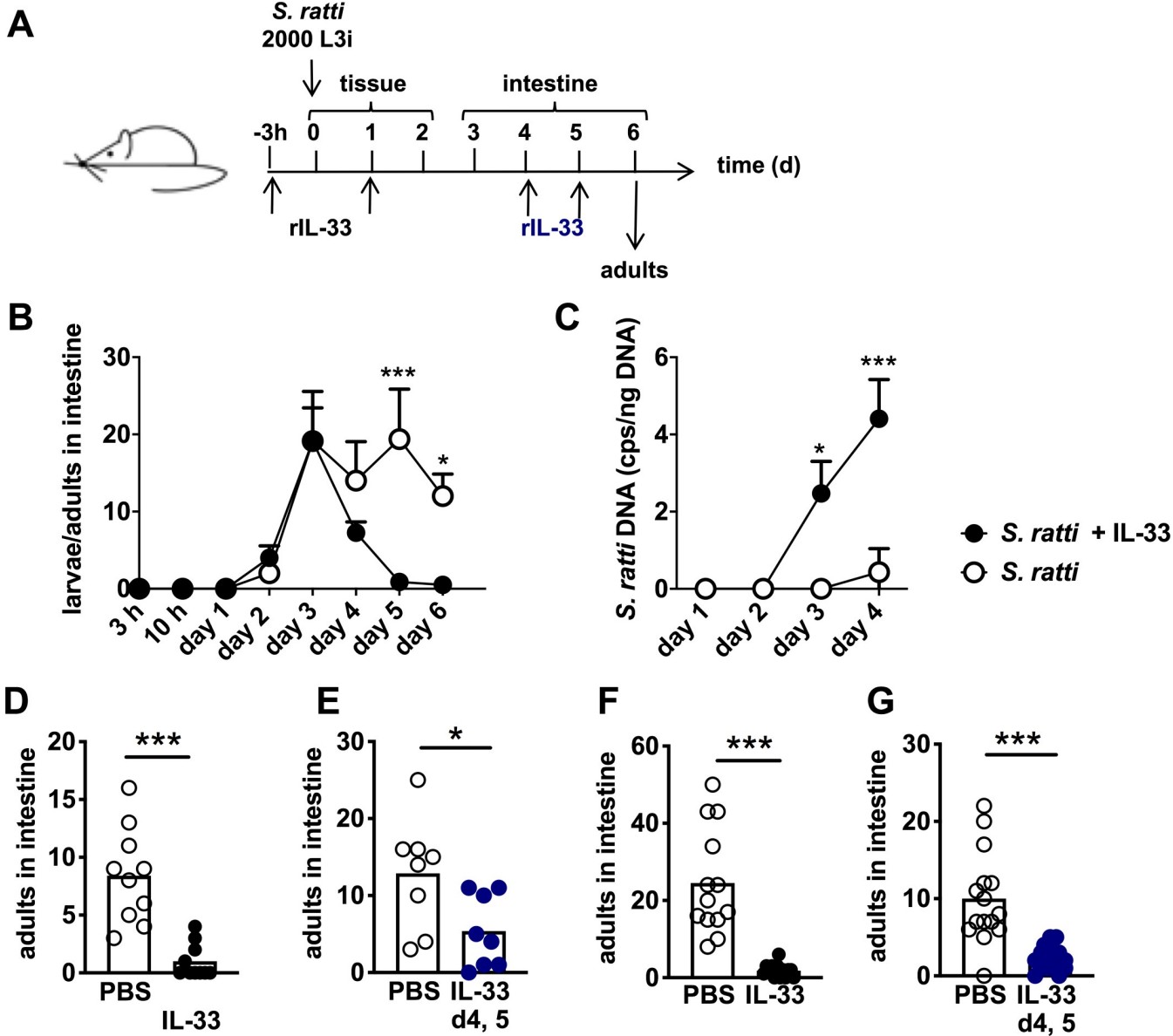

**Fig 2. IL-33 reduces intestinal *S. ratti* burden independent of the tissue migration phase. (A)** Experimental procedure: BALB/c mice were treated i.p. (**B,C, F, and G**) or i.n. (**D and E**) with PBS (open circles) or 1 µg rec. IL-33 (closed circles). Treatment was performed either 3 h before and 24 h after *S. ratti* infection (black circles, **B,C,D, and F**) or after the tissue migration phase i.e. 4 days and 5 days after *S. ratti* infection (blue circles, **E and G**). **(B)** Mice were sacrificed at the indicated time points and L3, L4 and adults in the intestine were counted and **(C)** *S. ratti*-derived DNA in the feces was quantified by qPCR at the indicated time points. Shown are the combined results of 2 independent experiments (n = 4 per experiment, group, and time point), symbols show the mean and error bars indicate SEM **(D-G)**. Adults in the intestine were counted at day 6 p.i. Shown are combined results from 2–3 independent experiments (n = 3–5 each), each symbol represents an individual mouse, bars show the mean and asterisks indicate statistically significant difference of the means (**B and C**: two-way ANOVA, **D-G**: students t-test).

(Fig 2F) and also i.p. application of IL-33 after completion of the tissue migration phase at day 4 and 5 p.i., reduced the intestinal parasite burdens on the next day (Fig 2G).

Taken together these results show that endogenous IL-33 which is naturally produced and/ or released during *S. ratti* infection promoted eradication of parasites from the intestine as blockade of endogenous IL-33 increased and stabilization of endogenous IL-33 decreased parasite burden. Exogenous application of rec. IL-33 phenocopied stabilisation of endogenous IL-

33. Thereby rec. IL-33 promoted *S. ratti* eradication (i) directly in the intestine, (ii) independent of the tissue migration phase and (iii) whether IL-33 was delivered to the lung or systemically.

## IL-33 mediates accelerated intestinal expulsion of *S. ratti* parasites independent of adaptive immunity via mucosal mast cells

Expulsion of *S. ratti* from the intestine is promoted by basophils [10] but predominantly executed by mucosal mast cells [12]. Therefore, we analyzed mast cell activation by quantification of mouse mast cell protease 1 (mMCPT-1) in the serum that is released by de-granulating mucosal mast cells [32] (Fig 3A). Although this method does not strictly distinguish between increased mucosal mast cell degranulation and increased numbers of degranulating mucosal mast cells, it reflects mast cell activation as the net-effect of mastocytosis and degranulation *in vivo*. *S. ratti* infection induced detectable mast cell activation by day 6 p.i. (Fig 3A black bars and asterisk), as we have shown before [15,16]. *S. ratti*-infected and IL-33-treated mice displayed significant elevation in mMCPT-1 serum concentrations already by day 1 p.i. and maximal mast cell activation by day 3 p.i., i.e. the time point of L3 arrival in the intestine (Fig 3A red bars and asterisks).

IL-33 treatment alone, in the absence of *S. ratti* infection, also led to rapid mast cell activation as soon as 3–10 hours post treatment, and this was sustained for 6 days i.e. 5 days after the second IL-33 injection (Fig 3A blue bars and asterisks). Of note, systemic i.p. and "lung-targeted" i.n. application of IL-33 induced mucosal mast cell activation to the same extent (S2C Fig). Comparison of IL-33-treatment in the absence and presence of *S. ratti* infection revealed even higher levels of IL-33-induced mast cell degranulation in non-infected mice, a trend that reached statistical significance at 10 hours and 4 days post treatment/infection compared to IL-33-treated *S. ratti*-infected mice (Fig 3A green asterisks).

To directly test the impact of endogenous IL-33 on mast cell activation, we next quantified mMCPT-1 concentration in the serum of *S. ratti*-infected mice that received HpARI or CCP1/2. Blockade of IL-33 via HpARI significantly reduced mast cell degranulation that was induced by *S. ratti* infection day 6 p.i. (Fig 3B), while stabilization of IL-33 via CCP1/2 increased mast cell degranulation, thus phenocopying treatment with rec. IL-33 (Fig 3C).

To provide a causal link between the IL-33-induced increased mast cell activation and the reduced worm burdens observed in rec. IL-33-treated and *S. ratti*-infected mice, we employed SCID mice that lack T and B cells and kit-independent mast cell-deficient Cpa3[Cre] mice [33]. IL-33-treatment reduced intestinal parasite burdens by day 6 p.i. in SCID mice i.e. in the absence of adaptive immunity but in the presence of mast cells (Fig 3D). By contrast, IL-33-treatment reduced intestinal parasite burdens day 6 p.i. selectively in mast cell-competent mice but not in the mast cell-deficient littermates (Fig 3E). These results show that IL-33-treatment prevented intestinal embedding of *S. ratti* by rapid mast cell activation within the first days of infection, independent of adaptive immunity but strictly dependent on mast cells.

## IL-33 mediates accelerated intestinal expulsion of *S. ratti* parasites independent of basophilic, eosinophilic and neutrophilic granulocytes and dependent of IL-9 receptor signalling

To analyse the contribution of additional innate effector cells in mediating the rapid IL-33-driven *S. ratti* expulsion from the intestine we focused on basophilic, eosinophilic, and neutrophilic granulocytes that contribute to the control of *Strongyloides* infection alongside mast cells [4]. Basophils are dispensable for the control of tissue-migrating *S. ratti* larvae but contribute to early anti-*Strongyloides* immunity in the intestine [10]. Basophil-deficient Mcpt8[Cre]

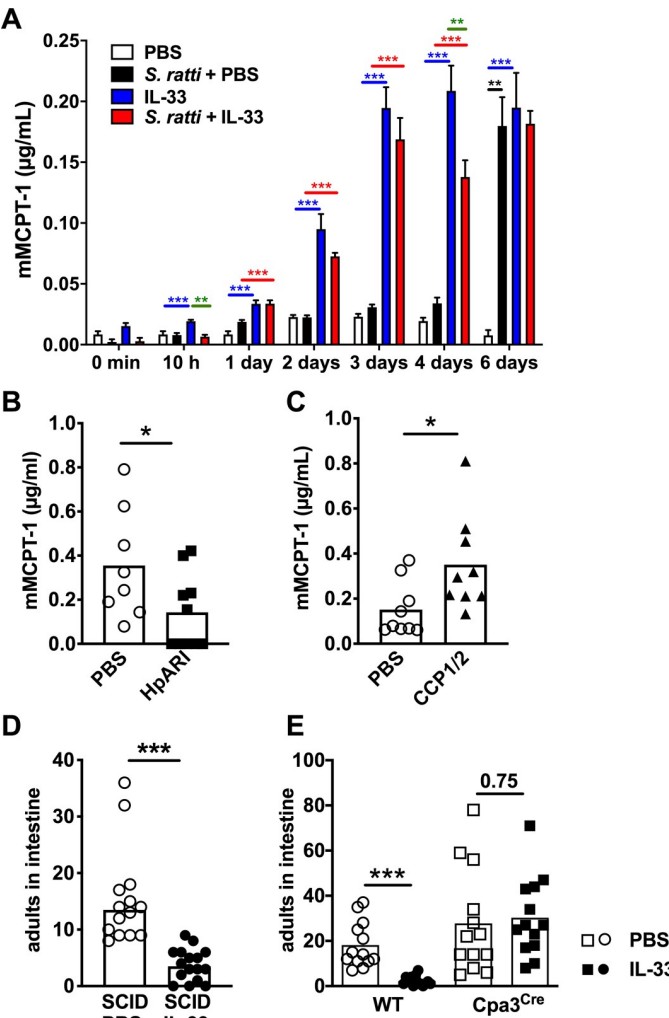

**Fig 3. IL-33 treatment induces rapid mast cell activation that mediates accelerated intestinal parasite expulsion.** **(A)** BALB/c mice were either left uninfected (white and blue bars) or infected with 2000 *S. ratti* L3 s.c. (black and red bars). Mice received PBS (white bars, black bars) or 1 μg rec. IL-33 (blue bars, red bars) either 3 h before and 24 h post *S. ratti* infection or mock infection. mMCPT-1 in the serum was quantified by ELISA at the indicated time points. Graph shows combined results of 2–3 independent experiments (n = 3–5 per experiment and time point, n = 2 for PBS control), bars indicate the mean and error bar show SEM. Asterisks indicate statistically significant differences of the mean of *S. ratti* + PBS to PBS in black; IL-33 to PBS in red: *S. ratti* + IL-33 to *S. ratti* + PBS in blue and *S. ratti* + IL-33 to IL-33 in green (one-way ANOVA performed for the 4 groups at each time point separately). **(B and C)** BALB/c mice received PBS (open circles) or **(B)** 5 μg rec. HpARI (closed squares) or **(C)** 5 μg rec. CCP1/2 (closed triangles) i. p. 3 h before and 24 h post s.c. infection with 2000 *S. ratti* L3. mMCPT-1 in the serum was quantified by ELISA at the indicated time points. **(D)** SCID mice or **(E)** Cpa3$^{Cre}$ mice (squares) and wildtype littermates (circles) were treated with 1 μg rec. IL-33 (closed symbols) or with PBS (open symbols) 3 h before and 24 h post *S. ratti* infection. Mice were sacrificed day 6 p.i. to count adults in the intestine. Graphs show the combined results of 2–4 independent experiments (n ≥ 3–5 per experiment and group), each symbol represents an individual mouse, bars show the mean and asterisks indicate statistically significant differences of the mean, numbers indicate p value (**B,C, and E:** students t-test **D:** Mann-Whitney test).

mice [34] displayed higher intestinal parasite burdens than the basophil-competent wildtype littermates (Fig 4A left), as expected [10]. Nevertheless, rec. IL-33-treatment reduced intestinal parasite burdens in basophil-deficient and basophil-competent mice to the same level. Reduced intestinal parasite burdens were correlated with increased mast cell activation

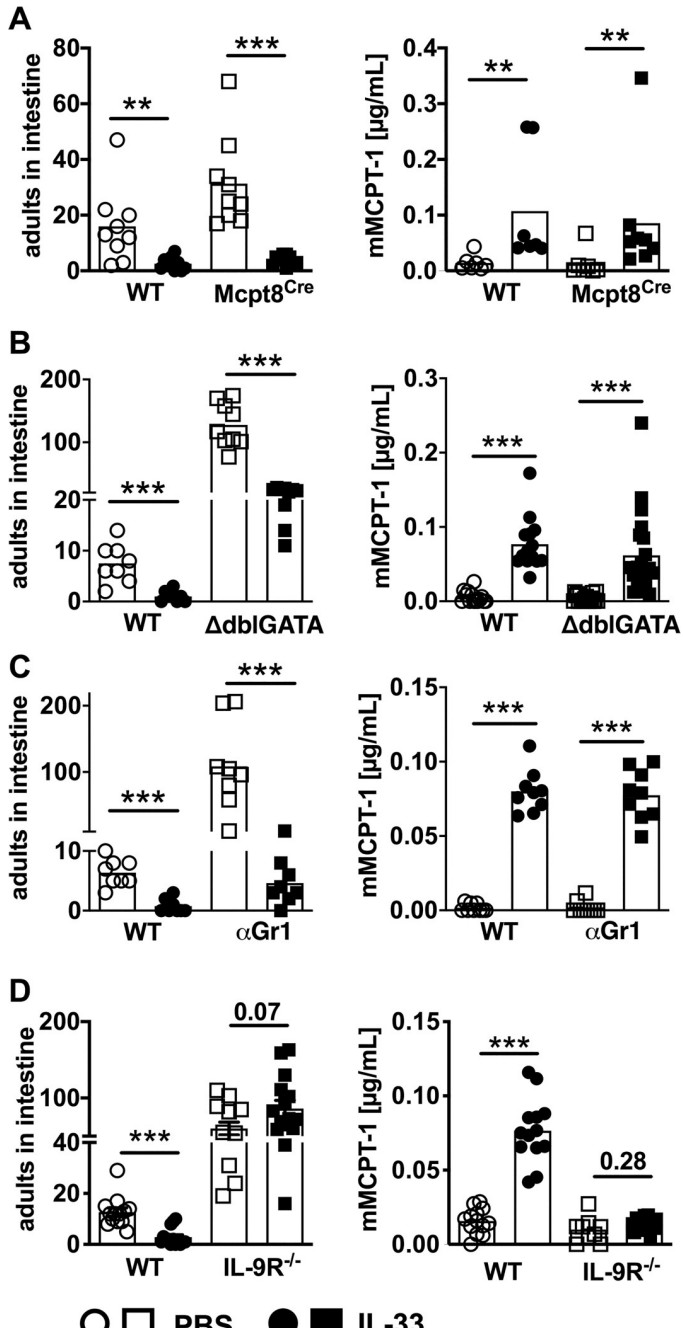

**Fig 4. IL-33 mediates accelerated intestinal expulsion of *S. ratti* parasites independent of basophilic, eosinophilic and neutrophilic granulocytes but dependent of IL-9. (A)** BALB/c Mcpt8^Cre mice (squares) and non-transgenic littermates (circles) or **(B)** ΔdblGATA mice (squares) and co-housed BALB/c mice (circles) or **(C)** BALB/c mice treated with anti Gr-1 mAb (depletion protocol and control is shown in S3 Fig) (squares) or isotype control, and **(D)** BALB/c IL-9 receptor-deficient mice (squares) and co-housed BALB/c mice (circles) were treated i.p. with 1 μg of IL-33 (closed symbols) or with PBS (open symbols) 3 h before and 24 h post *S. ratti* infection. mMCPT-1 concentration in the sera was quantified day 2 p.i. and parasitic adults in the intestine counted day 6 p.i. Graphs show combined results from 2–4 independent experiments (n = 3–8 per experiment and group). Each symbol represents an individual mouse, bars show the mean, asterisks indicate statistically significant differences of the mean, numbers indicate p value (Mann-Whitney and students t-test).

indicated by elevated mMCPT-1 serum concentrations day 2 p.i., in the presence and absence of basophils (Fig 4A right).

Neutrophils and eosinophils predominantly mediate killing of tissue-migrating larvae, while their direct impact in intestinal expulsion is less clear [4,6,7]. In particular, eosinophils were associated with IL-33-mediated reduction of parasite load during *S. venezuelensis* infection [22]. ΔdblGATA mice that lack eosinophilic granulocytes displayed elevated intestinal parasite burdens compared to wildtype BALB/c mice. Despite their almost ten-fold increased numbers of parasitic adults in the intestine, ΔdblGATA mice responded to rec. IL-33 application with a significant reduction of intestinal parasite load that was accompanied by increased mast cell activation (Fig 4B).

Depletion of all granulocytes including neutrophils by application of an anti-Gr-1 mAb *in vivo* one day before and one day after *S. ratti* infection (S3 Fig) phenocopied ΔdblGATA mice. Intestinal parasite burdens were higher in granulocyte-depleted mice compared to non-depleted mice but were still significantly reduced by additional treatment with rec. IL-33 in the context of accelerated mast cell activation (Fig 4C). These combined results show that out of all innate effector cell populations contributing to *S. ratti* eradication, IL-33 selectively targeted mast cells to accelerate intestinal expulsion.

Since mast cells express the IL-33 receptor ST2 [35] and can be activated by IL-33 [36,37], it is conceivable that they responded directly to IL-33. However, we have previously shown that the rapid early mast cell activation during *S. ratti* infection depended on functional IL-9 receptor signalling [14] and that IL-9 production during *S. ratti* infection was actively antagonized by the parasite as a strategy of immune evasion [16,17]. Therefore, we tested a putative contribution of IL-9 to accelerated mast cell activation and parasite expulsion observed in IL-33-treated mice. IL-9 receptor-deficient (IL-9R$^{-/-}$) mice displayed increased intestinal parasite burdens (Fig 4D), as expected [14]. Treatment with rec. IL-33 did not reduce parasite numbers and did not increase the early mast cell activation day 2 p.i. (Fig 4D). This was not due to the general elevation of intestinal parasite burdens in IL-9R$^{-/-}$ mice because similarly elevated parasite burdens were readily reduced by rec. IL-33 treatment in basophil- eosinophil- and granulocyte-deficient mice (Fig 4A–4C).

## IL-33-expanded ILC2 contribute to rapid mast cell activation and expulsion of *S. ratti* from the intestine

Since the dominant innate source of IL-9 are ILC2s [38], we compared BALB/c RAG$^{-/-}$ mice that lack adaptive immunity but are ILC-competent to BALB/c RAG$^{-/-}$γc$^{-/-}$ mice which are additionally ILC-deficient (Fig 5). ILC2s were defined as lineage$^-$ CD127$^+$ Eomes$^-$ RORγt$^-$ GATA3$^+$ cells (S4 Fig). ILC2s were completely absent in lung, spleen and peritoneum of both, untreated and IL-33-treated RAG$^{-/-}$γc$^{-/-}$ mice, while IL-33 treatment of BALB/c RAG$^{-/-}$ mice expanded ILC2s (Fig 5A and 5B). IL-33 treatment reduced parasite burdens in the context of accelerated mast cell activation in BALB/c RAG$^{-/-}$ mice (Fig 5C and 5D circles) thereby reiterating our previous results recorded in SCID mice (Fig 3D). Additional absence of ILCs in BALB/c RAG$^{-/-}$γc$^{-/-}$ mice increased intestinal parasite burdens, thus highlighting the general contribution of ILC2s to the anti-*Strongyloides* immune response (Fig 5C squares). IL-33 treatment did not induce detectable mast cell activation in BALB/c RAG$^{-/-}$γc$^{-/-}$ mice at day 3 or day 6 p.i., and parasite burdens were not significantly reduced in IL-33-treated BALB/c RAG$^{-/-}$γc$^{-/-}$ mice (Fig 5C and 5D). These results strongly suggest that ILC2s contributed to the IL-33-induced acceleration of mast cell activation and subsequent improved expulsion of *S. ratti* from the intestine. It should be noted that we recorded a non-significant trend (p = 0.075) to reduced parasite burdens in IL-33-treated BALB/c RAG$^{-/-}$γc$^{-/-}$ mice (i.e. in the complete

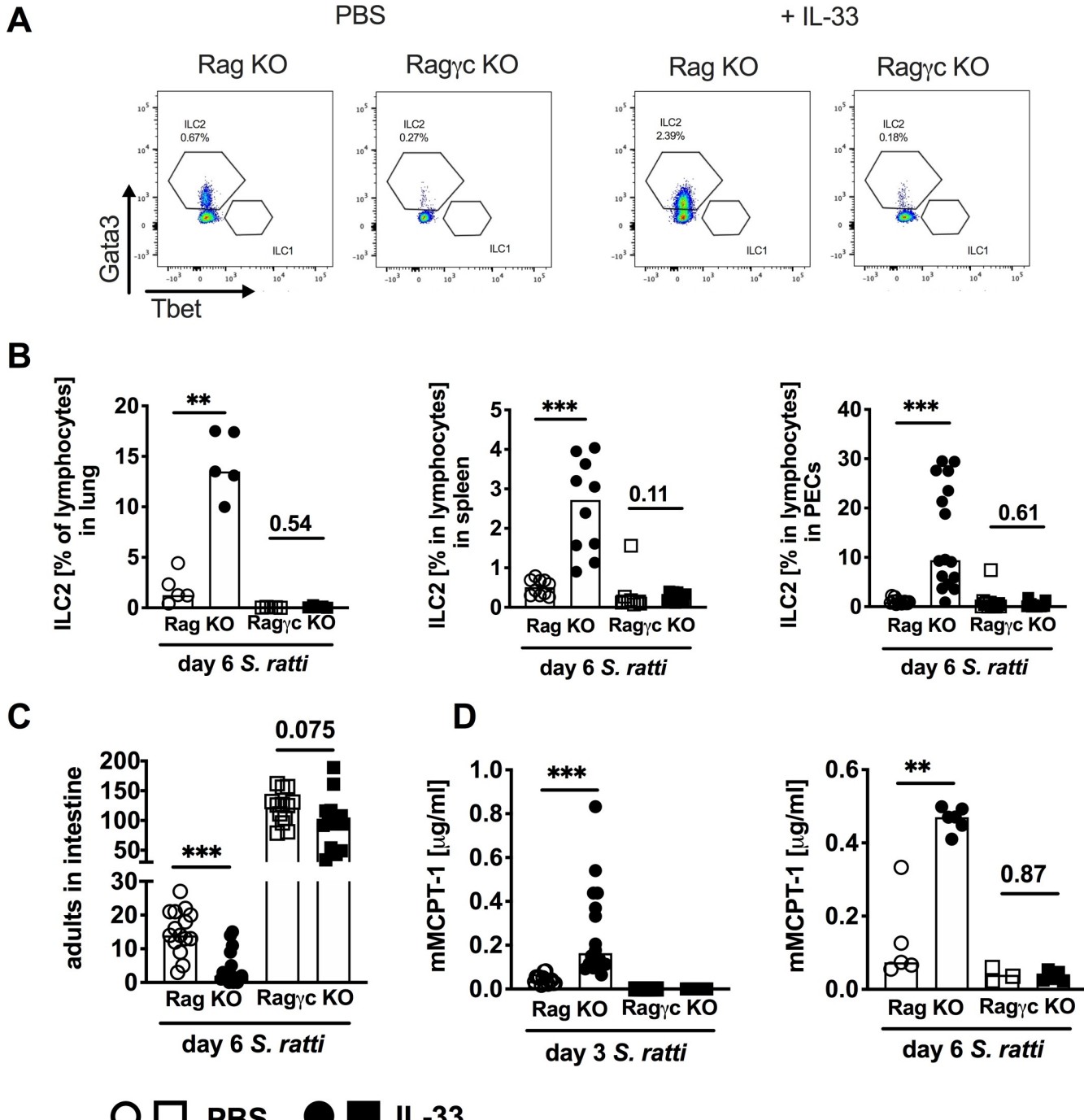

**Fig 5. ILC promote IL-33-mediated *S. ratti* expulsion from the intestine. (A-D)** BALB/c RAG<sup>-/-</sup> mice (circles) and BALB/c RAG<sup>-/-</sup> γc<sup>-/-</sup> mice (squares) were treated i.p. with 1 μg of IL-33 (closed symbols) or with PBS (open symbols) 3 h before and 24 h post *S. ratti* infection. **(A)** Representative dot blots showing frequency of ILC2 in spleens of BALB/c RAG<sup>-/-</sup> mice or BALB/c RAG<sup>-/-</sup> γc<sup>-/-</sup> mice with or without IL-33 treatment. Cells were measured using an LSRII Cytometer (BD, Germany) and analyzed by FlowJo software. **(B)** Frequencies of lung, spleen and PEC cells day 6 p.i. Graphs show combined results from 1 (lung), 2 (spleen) or 3 (PEC) independent experiments. **(C)** Parasitic adults in the intestine were counted 6 days p.i. and **(D)** mMCPT-1 concentration in the sera was quantified at indicated time points p.i. Graphs show combined results from 3 independent experiments (n = 4 per experiment and group) or 1 experiment (day 6 mMCPT-1). Each symbol represents an individual mouse, bars show the mean, asterisks indicate statistically significant differences of the mean, numbers indicate p value (Mann-Whitney test (**B and D**) and students t-test (**C**).

absence of ILC2), with no evidence of mast cell activation measured by serum mMCPT-1. This may reflect the existence of an alternative ILC- and mast cell-independent axis of IL-33-mediated immunity that compensates for the absent ILC and is visualised in these extreme models of effector cell deficiency.

## Discussion

The tissue-derived alarmin cytokine IL-33 plays a central role in the initiation of rapid and appropriate immune responses to parasitic helminths. Taking advantage of the IL-33 inhibitor HpARI [30], and a truncation mutant of HpARI (CCP1/2) which amplifies IL-33 responses *in vivo* [31], we tested the impact of endogenous IL-33 on the early immune response to *S. ratti*.

The activity of HpARI is due to specific and high-affinity binding directly to the IL-33 cytokine. In doing so, HpARI blocks interaction of IL-33 with its receptor, ST2. Furthermore, HpARI binds to genomic DNA: this dual DNA/IL-33 binding allows HpARI to tether IL-33 within necrotic cells, preventing release of the cytokine [30]. The CCP1/2 truncation mutant of HpARI was shown to amplify rather than suppress IL-33 responses *in vitro* and *in vivo*, in a range of cytokine-, allergen-, and infection-driven systems [31]. CCP1/2 retains the ability to bind to DNA and IL-33, however cannot block the interaction of the cytokine with its receptor. This non-blocking binding appears to stabilise the cytokine in its active form, extending its half-life *in vivo* and amplifying endogenous IL-33 responses.

Application of HpARI prior to *S. ratti* infection increased the intestinal parasite numbers day 6 p.i. in the context of reduced mast cell activation, while either stabilization of endogenous IL-33 by application of CCP1/2 or direct application of rec. IL-33 reciprocally reduced parasite burdens in the context of increased mast cell activation. Using these IL-33-specific enhancers and inhibitors, we thereby establish a biologic function for endogenous IL-33 that is released during *S. ratti* infection in mast cell-mediated parasite control.

Our further experiments using application of rec. IL-33 strongly suggest that the IL-33-mediated anti-*S. ratti* immunity was not established during tissue migration, but rather during the later intestinal phase. Kinetic analyses of intestinal parasite burdens revealed that while similar parasite numbers arrived in the intestine of rec. IL-33 treated mice day 3 p.i., IL-33 treatment prevented successful embedding in the intestine. Application of rec. IL-33 at days 4 and 5 p.i., that is after the tissue migration phase, still reduced intestinal parasite burdens, thus excluding a role for IL-33 in eradicating the tissue migrating L3. Furthermore, deficiency of either eosinophils or granulocytes that have been shown to mediate killing of migrating *Strongyloides* larvae in the tissues [5–9] did not abrogate the IL-33-mediated reduction of intestinal parasite burdens. Even though the initial parasite numbers in the intestine increased ten-fold in eosinophil-deficient ΔdblGATA and granulocyte-depleted mice, IL-33 treatment still caused a significant reduction.

Our results are in contrast with two studies suggesting that IL-33 promoted eosinophil-dependent eradication of tissue-migrating *S. venezuelensis* larvae in C57BL/6 mice via ILC2-derived IL-5 [22,39]. The authors reported that IL-33 deficiency increased *S. venezuelensis* faecal egg release at day 8 p.i. and simultaneously abrogated lung ILC2 expansion, IL-5 production and lung eosinophilia, while i.n. application of recombinant IL-33 rescued these features in IL-33[-/-] mice [22]. A previous *S. venezuelensis* infection rendered C57BL/6 mice also more resistant to subsequent *N. brasiliensis* infection, reducing intestinal parasite burden in an IL-33-, ILC-, IL-5- and eosinophil-dependent manner but independent of CD4[+] T cells [39]. Direct evidence for a role of IL-33 in enhancing Th2-mediated attack of migrating parasitic nematodes in the lung arises from two other studies: IL-33 release triggered by house dust mite allergen administration reduced numbers of *Ascaris lumbricoides* larvae in the lung

during subsequent infection [40], while *H. polygyrus* infection-induced IL-33 reduced *N. brasiliensis* larvae counts in the lungs of C57BL/6 mice [41]. Reduction of migrating larvae was dependent on IL-5 and eosinophils, but also on a functional adaptive immune system in both studies.

In the current study, we rule out a comparable impact of IL-5-activated eosinophils, since administration of rec. IL-33 reduced intestinal *S. ratti* parasite burdens in eosinophil-deficient mice and in granulocyte-depleted mice to the same extent as in wildtype mice. This discrepancy may reflect the different migration routes that are almost exclusively the lung route for *S. venezuelensis*, *A. lumbricoides* and *N. brasiliensis* while *S. ratti* also migrates via skin and muscle tissues directly to the mouth. Moreover, the functions of IL-33 in enhancing adaptive type 2 immunity and thereby improving larval attack in secondary infections will be different from the rapid and innate function IL-33 exerts during the first week of infection that we study here. Finally, most of the cited studies used C57BL/6 mice [22,39,41] while we used BALB/c mice in our current study. As we have reported mouse strain-specific differences in the regulation of anti-*S. ratti* immune response in BALB/c and C57BL/6 mice previously [16] it is conceivable that the role of eosinophils in mediating IL-33 triggered anti-helminth immunity may be different in C57BL/6 mice.

Our combined results regarding *S. ratti*-infected BALB/c mice rather support a model where IL-33 administration triggered rapid activation of mucosal mast cells in the intestine, thereby preventing successful embedding of arriving parasites because (i) rec. IL-33 administration led to detectable mucosal mast cell degranulation within 3 hours that peaked at 3 days, i.e. the timepoint when larvae reach the intestine, (ii) newly arriving *S. ratti* larvae did not attach to the intestine in IL-33-treated mice but were detected in faeces, and (iii) IL-33 reduced intestinal parasite burdens only in the presence of mast cells while neither eosinophils, neutrophils, nor basophils that also contribute to controlling intestinal parasite burdens [10,11], were needed to translate IL-33 application into reduced intestinal parasite burdens.

Although mast cells express the IL-33 receptor ST2 [18,36] and direct induction of cytokine production but not degranulation of mast cells by IL-33 was demonstrated *in vitro* [36,37], we provide evidence that mast cells were predominantly activated indirectly via ILC2s and IL-9. In line with previous studies [42–44] we observed that treatment with rec. IL-33 induced ILC2 expansion. Absence of either ILC2s in BALB/c RAG$^{-/-}$γc$^{-/-}$ mice and/or absence of functional IL-9 receptor signalling in IL-9R-deficient mice abrogated both, IL-33-mediated reduction of intestinal parasite burdens and activation of mast cells. These findings reveal that ILC2s and IL-9 acted upstream of the mast cell activation in a non-redundant manner.

We did not attempt to identify the cellular source(s) of IL-9 in this study, as we have previously shown that T cells and lineage-negative cells that most likely represent ILC2 produce IL-9 during *S. ratti* infection [16]. A central contribution of IL-9-producing Th9 cells in eradication of intestinal helminths was demonstrated in *N. brasiliensis* infected mice [45] and may be important at later time points of *S. ratti* infection as well. However, the rapid induction of IL-33-mediated mast cell activation reported in the current study was established in SCID and RAG$^{-/-}$ mice independently of T cells and thus independently of T cell-derived IL-9. As ILC2s are the major innate producers of IL-9 [38], our results are in line with the following chain of events illustrated in Fig 6: (i) migrating *S. ratti* induce release and/or production of IL-33, (ii) IL-33 activates ILC2 that produce IL-9 which further activates ILC2, (iii) IL-9 directly and/or indirectly activates mucosal mast cells to degranulate and promote intestinal immunity (iv) independently of T and B cells, eosinophils, basophils or neutrophils, (v) resulting in newly arriving larvae encountering a hostile environment in the intestine and failing to embed into the mucosa.

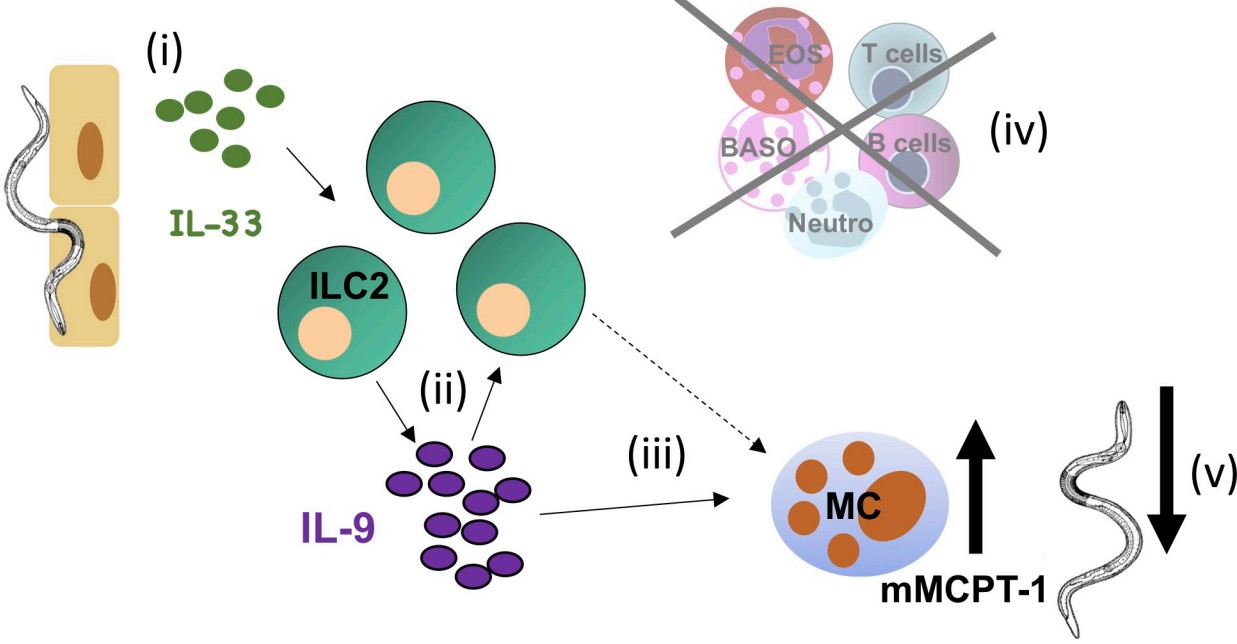

**Fig 6. Role of IL-33 during *S. ratti* infection.** The cartoon illustrates the proposed functions of IL-33 during *S. ratti* infection. (i) Migrating *S. ratti* larvae induce IL-33 that (ii) activates ILC2 to produce IL-9 that further activates ILC2, (iii) IL-9 directly and/or indirectly activates mucosal mast cells (iv) independently of T and B cells, eosinophils, basophils or neutrophils, (v) to promote ejection of *S. ratti* from the intestine.

Our working hypothesis (Fig 6) is supported by several studies. (i) Release of IL-33 protein and upregulation of IL-33 transcription is induced by a range of damaging stimuli to the organs that *S. ratti* migrates through, such as tape stripping-induced mechanical damage of the skin [46], protease-containing allergen damage to the lung [47], and migration of *S. venezuelensis*, *N. brasiliensis* or *T. muris* parasites through the lung [21,22] or intestine [23]. (ii) IL-33, in concert with other alarmin cytokines such as IL-25 was shown to mediate expansion of ILC2s [42–44] that are the dominant source of innate IL-9 [21,38]. (iii) IL-9 promotes mucosal mast cell activation [14] and (v) activated mast cells mediate *S. ratti* expulsion from the intestine [12].

In summary, by treating mice with IL-33 inhibitors and stabilizers in this study, we provide direct evidence that endogenous IL-33 that is produced naturally during *S. ratti* infection promotes mast cell activation and intestinal parasite control in a non-redundant manner. Using application of exogenous rec. IL-33 allowed us to sensitively dissect the role of IL-33 in this system. Paired with the experiments assessing endogenous IL-33 responses we highlight the importance of an IL-33 –ILC2 –IL-9 –mast cell axis in the initiation of a type 2 immune response to tissue-migrating parasites, leading to rapid generation of a hostile environment in the intestine that facilitates parasite expulsion. A deeper understanding of the specific immune pathways that lead to parasite expulsion may facilitate development of more potent treatments for chronic helminth infections.

## Material and methods

### Ethics statement

Animal experiments were conducted in agreement with the German animal protection law and experimental protocols were approved by Federal Health Authorities of the State of

Hamburg (permission-numbers 55/13, 111/16 and A029/18). All mice were bred in the animal facility of the BNITM and kept in individually ventilated cages under specific pathogen-free conditions. Mice were sacrificed by an overdosed $CO_2$ narcosis followed by cervical dislocation in accordance with the German animal protection law.

## Mice

BALB/c RAG$^{-/-}$, BALB/c RAG$^{-/-}$γc$^{-/-}$, IL-9R-deficient BALB/c mice [48], mast cell-deficient BALB/c Cpa3$^{Cre}$ mice [33], basophil-deficient Mcpt8$^{Cre}$ mice [34], and eosinophil-deficient ΔdblGATA mice have been described before. For all experiments, male and female mice were used at 7 to 10 weeks of age, but experimental groups were matched for sex and age with maximally 7 days variance.

## Parasites and Infection

The *S. ratti* cycle was maintained in Wistar rats and infections were performed by s.c. infection of 2000 L3 in in 30 μl PBS into the hind footpad of mice as described [49,50]. Parasite burden in tissue and intestine and quantification of the *S. ratti* 28S RNA-coding DNA in the faeces of infected mice was performed as described [50]. For IL-33 treatment 1μg rec IL-33 (Biolegend, Catalogue Nr. 580508) was applied either i.p. in 200 μl PBS or i.n. in 20 μl PBS either 3 h before and 1 day after *S. ratti* infection or at days 4 and 5 of *S. ratti* infection. Some mice received 350 μg anti Gr-1 (clone RB6-8C5) at -1 and day 1 of *S. ratti* infection. Depletion of cells was verified by flow cytometry (S3 Fig).

## HpARI and CCP1/2

The HpARI and CCP1/2 proteins were produced in Expi293 cells and purified as previously described [30,31]. Briefly, Expi293 cells (ThermoFisher) were transfected with pSecTAG2A plasmids (Thermofisher Scientific) containing inserts encoding HpARI or CCP1/2 (with C-terminal Myc and 6-His tags), using Expifectamine transfection and enhancer reagents (Thermofisher Scientific) and following manufacturer´s instructions. Culture supernatants were collected 7 days after transfection, and tagged proteins purified by sequential Nickel column purifications on HisTRAP excel and HiTRAP chelating HP columns (Merck). Proteins were eluted from columns using an imidazole gradient, and fractions containing proteins of interest were pooled, dialysed to PBS, filter sterilised and protein concentration calculated by A280 measurements.

## Flow Cytometry

Lung, spleen and peritoneal cavity cells (PECs) were isolated and single cell suspension prepared. Single cells were isolated from the lung as described before [14]. For surface staining, 3–5 x 10$^6$ cells were stained for 25 minutes at 4˚C with Biotin-labeled (lineage cocktail) targeting mouse CD11b (clone M1/70), CD8 (clone 53–6.7), CD19 (clone 6D5), CD11c (clone N418), CD3 (clone 17A2), TCRβ (clone H57-97), TCRγδ (Clone GL3), Gr-1 (clone RB-8C5), CD5 (clone 53–7.3), CD49b (clone DX5), TER-119 (clone TER-119) and NK1.1 (clone PK136), BV510-labeled anti-mouse CD4 antibody (clone RM4-5), AF700-labeled anti-mouse CD45 antibody (clone 30-F11), PE-Cy7-labeled anti-mouse CD90.2 antibody (Clone30-H12) and BV421-labeled anti-mouse CD127 antibody (clone A7R34). Subsequently, cells were washed and stained for 15 minutes at 4˚C with PerCP Cy5.5-labeled Streptavidin. For intracellular staining, first cells were fixed and permeabilized using the Thermofisher Scientific Foxp3/Transcription factor staining buffer set according to the manufacturer's protocol.

Intracellular staining was performed using the following antibodies: AF488-labeled anti-mouse GATA3 antibody (clone L50-823, from BD), PE-labelled anti-mouse Eomes antibody (clone Dan11mag), APC-labelled anti-mouse RorγT antibody (clone Q31-378, from BD) and PE/Dazzle594-labeled anti-mouse T-bet antibody (clone 4B10). Antibodies were purchased from BioLegend or Thermofisher Scientific if not stated otherwise. Samples were analysed on a LSRII (Becton Dickinson) using FlowJo software (TreeStar). The gating strategy for ILC2 is shown in S4 Fig.

## Mast cell activation

Blood was collected from infected mice at the indicated time points and allowed to coagulate for 1 h at room temperature (RT). Serum was collected after centrifugation (10.000 x g) for 10 min at RT. mMCPT-1 concentration was quantified using MCPT-1 Ready-SET-Go kit (Thermofisher Scientific) according to the manufacturer's recommendations.

## Statistical analysis

All data were assessed for normality and groups were compared by using Student's *t*-test, one-way ANOVA, two-way ANOVA (parametric), Mann Whitney-U test, or Kruskal-Wallis test (non-parametric) using GraphPad Prism software (San Diego) as indicated in the figure legends. P values of $\leq 0.05$ were considered to indicate statistical significance. Asterisk indicate statistically significant differences [*] $p < 0.05$; [**] $p < 0,01$; [***] $p < 0,001$. The numerical data used to generate Figs 1–5 and S1–S3 Figs are provided in the supplementary files (S1–S8 Data).

## Supporting information

**S1 Fig. (related to Fig 1). IL-33 release by tissue explants** BALB/c mice were left naïve (open circles) or s.c. infected with 2000 *S. ratti* L3 (closed circles). Mice were sacrificed day 2 and day 6 p.i. and lungs and small intestine prepared. Tissue explants corresponding to half a lung (ca 100 mg) or one tenth of the small intestine (ca 150 mg) were weighed, placed in 48 well plates in 250 μl serum-free RPMI 1640 medium supplemented with 100 U/ml Penicillin/Streptavidin and cOmplete Protease Inhibitor cocktail (Roche) and incubated for 24 h at 37°C. IL-33 in the SN was quantified using an IL-33 ELISA Kit (Invitrogen by Thermfisher Scientific) according to the manufacturer's recommendation and normalized to the weight of the explants. Graphs show the combined results of 1 (day 2) or 3 independent experiments (n = 2–8 per group and experiment). Each symbol represents an individual mouse, bars show the mean, number indicate the p value and asterisk indicate statistically significant differences between groups (Mann-Whitney test).
(PDF)

**S2 Fig. (related to Fig 2). Intranasal application of IL-33 results in systemic elevation of IL-33 concentration and mucosal mast cell activation (A)** Experimental procedure: BALB/c mice were treated i.n. (open circles) or i.p. (closed circles) with 1 μg rec. IL-33 3 h before and 24 h post *S. ratti* infection. Serum samples were taken at the indicated time points and **(B)** IL-33 and **(C)** mMCPT-1 concentration in the sera were quantified pre-treatment (0), 3 h, 1 and 3 days after treatment by ELISA. Shown are combined results from 2 independent experiments (n = 3–5; pre-treatment n = 2 per experiment and group) each symbol represents an individual mouse, bars show the mean and asterisk indicate statistically significant difference of the means compared to pre-treatment (one-way ANOVA).
(PDF)

**S3 Fig. (related to Fig 4). Depletion of Gr-1$^+$ cells (A)** Experimental procedure: BALB/c mice received i.p. 350µg anti-Gr-1 mAb (clone RB6-8C5, squares) or isotype control (circles) one day before and one day after *S. ratti* infection. Mice were additionally treated with 1 µg of IL-33 (closed symbols) or with PBS (open symbols) 3 h before and 24 h post *S. ratti* infection. Frequency of Gr-1$^+$ CD11b$^+$ cells in the leukocyte gate of PBS were measured by flow cytometry at day 1 p.i. To this end cells were stained with anti-mouse/human CD11b-PerCP-Cy5.5 (M1/70) and anti-mouse Gr-1-BV421 (RB6-8C5) (both BioLegend, Germany), measured with an LSRII Cytometer (BD, Germany) and analyzed by FlowJo software. **(B)** Representative dot blots and **(C)** combined results of 2 independent experiments (n$\geq \geq$ 4 per experiment and group) showing frequency of granulocytes within PBL-leukocytes of the indicated groups are shown. Each symbol represents an individual mouse, bars represent the mean and asterisk indicate statistically significant differences of indicated groups (Kruskal-Wallis test with Dunn's post test).
(PDF)

**S4 Fig. (related to Fig 5). Gating of ILC2** ILC2 gating strategy is displayed for splenic cells isolated from a BALB/c RAG$^{-/-}$ mouse treated with 1 µg rec. IL-33. Cells were stained for 25 minutes at 4˚C with Biotin-labeled lineage cocktail (targeting mouse CD11b, CD8, CD19, CD11c, CD3, TCRβ, TCRγδ, Gr-1, CD5, CD49b, TER-119 and NK1.1) and PE-Cy7-labeled anti-mouse CD90.2 antibody and BV421-labeled anti-mouse CD127 antibody. Subsequently, cells were washed and stained for 15 minutes at 4˚C with PerCP Cy5.5-labeled Streptavidin. For intracellular staining, first cells were fixed and permeabilized using the Thermofisher Scientific Foxp3/Transcription factor staining buffer set according to the manufacturer's protocol. Intracellular staining was performed using the following antibodies: AF488-labeled anti-mouse GATA3 antibody, PE-labelled anti-mouse Eomes antibody, APC-labeled anti-mouse RorγT antibody, and PE/Dazzle594-labeled anti-mouse T-bet antibody. Cells were measured using an LSRII Cytometer (BD, Germany) and analyzed by FlowJo software.
(PDF)

**S1 Data. Prism File containing the numerical data used to generate Fig 1.**
(PZFX)

**S2 Data. Prism File containing the numerical data used to generate Fig 2.**
(PZFX)

**S3 Data. Prism File containing the numerical data used to generate Fig 3.**
(PZF)

**S4 Data. Prism File containing the numerical data used to generate Fig 4.**
(PZFX)

**S5 Data. Prism File containing the numerical data used to generate Fig 5.**
(PZFX)

**S6 Data. Prism File containing the numerical data used to generate S1 Fig.**
(PZFX)

**S7 Data. Prism File containing the numerical data used to generate S2 Fig.**
(PZFX)

**S8 Data. Prism File containing the numerical data used to generate S3 Fig.**
(PZF)

## Acknowledgments

We thank Marie-Luise Brunn for excellent technical assistance. We thank Prof. Jean-Christophe Renauld for sharing the IL-9 receptor-deficient mice, Prof. David Voehringer for sharing the basophil-deficient Mcpt8^Cre mice, Prof. Hans-Reimer Rodewald for sharing the Mast cell-deficient BALB/c Cpa3^Cre and Prof. Marc Hübner for providing ΔdblGATA mice.

## Author Contributions

**Conceptualization:** Jan-Eric Turner, Henry J. McSorley, Minka Breloer.

**Formal analysis:** Jana Meiners, Martina Reitz, Nikolas Rüdiger, Lennart Heepmann, Lena Rudolf, Wiebke Hartmann.

**Investigation:** Jana Meiners, Martina Reitz, Nikolas Rüdiger, Lennart Heepmann, Lena Rudolf, Wiebke Hartmann.

**Methodology:** Jana Meiners, Martina Reitz, Nikolas Rüdiger, Lennart Heepmann, Lena Rudolf, Wiebke Hartmann.

**Resources:** Jan-Eric Turner, Henry J. McSorley.

**Supervision:** Minka Breloer.

**Writing – original draft:** Minka Breloer.

**Writing – review & editing:** Jana Meiners, Jan-Eric Turner, Wiebke Hartmann, Henry J. McSorley, Minka Breloer.

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
