## [Decision Letter · Decision Letter 0]

2 Sep 2020

Dear PD Dr. Breloer,

Thank you very much for submitting your manuscript "IL-33 facilitates rapid expulsion of the parasitic nematode Strongyloides ratti from the intestine via ILC2- and IL-9-driven mast cell activation" for consideration at PLOS Pathogens. As with all papers reviewed by the journal, your manuscript was reviewed by members of the editorial board and by several independent reviewers. In light of the reviews (below this email), we would like to invite the resubmission of a significantly-revised version that takes into account the reviewers' comments.

We cannot make any decision about publication until we have seen the revised manuscript and your response to the reviewers' comments. Your revised manuscript is also likely to be sent to reviewers for further evaluation.

Sincerely,

Meera Goh Nair

Guest Editor

PLOS Pathogens

P'ng Loke

Section Editor

PLOS Pathogens

Kasturi Haldar

Editor-in-Chief

PLOS Pathogens

orcid.org/0000-0001-5065-158X

Michael Malim

Editor-in-Chief

PLOS Pathogens

orcid.org/0000-0002-7699-2064

Editor's comments:

Please address requests from Reviewers 1 and 2, specifically to provide data and results that further explore the significance of endogenous IL-33, and evaluate mast cell responses. Please also address Reviewers minor revision requests.

Reviewer's Responses to Questions

**Part I - Summary**

Reviewer #1: The authors have done a detailed study of the role of administered IL-33 on the primary response of BALB/c mice to the pathogen S. ratti. Studies focused on the role of endogenously produced IL-33 were more limited, as shown in some of the data in Fig. 1. As the authors state in the abstract: "Blockade of endogenous IL-33 using a helminth-derived IL-33 inhibitor elevated intestinal parasite burdens while stabilization of endogenous IL-33 or application of recombinant IL-33 reciprocally reduced intestinal parasite burdens." In all of the remaining figures, the approach was to treat BALB/c mice, or mutants on the BALB/c background, with exogenous IL-33, and to analyze the dependence of an appropriate response to IL-33 on various components of the primary immune response to S. ratti.

As a result, except for some of the data in Fig. 1, all studies are pharmacological analyses of the consequences of administered IL-33 on the BALB/c mouse response to S. ratti infection. The results of such studies show what the amounts of exogenous IL-33 used in the experiments can do, and the dependence on such results on particular immune cells. However, whether endogenous IL-33, in the amounts that can be produced in vivo, is able to mediate the same effects apparently has not yet been shown.

The following changes will render this report more useful in the growing literature regarding the primary immune response to S. ratti.

1) The authors should more clearly describe which experiments pertain to endogenous IL-33 (some of the data in Fig. 1) and which experiments use administered exogenous IL-33. They also should note explicitly that the latter experiments provide evidence of what endogenous IL-33 might do in this response, but do not prove that production of endogenous IL-33 has all of the effects shown for administered exogenous IL-33. More work addressing the role of endogenous IL-33 in this response would greatly increase the relevance of the study.

2) Since the results in Fig. 1 are the only data which focus on endogenous IL-33, more information about the natural products analyzed (i.e., the full length helminth-derived IL-33 inhibitor and the truncated version of that) should be provided. Specifically, how clear is the evidence that the effects of these agents, given in the amounts used in the authors' experiments, are solely attributable to effects on IL-33?

3) The authors should note more explicitly that their results pertain to the response to S. ratti in BALB/c mice, and such results may differ among other strains of mice.

Minor points:

1. The attached version of the manuscript has noted minor typographical problems, which need to be addressed.

Reviewer #2: This elegant study identifies that boosting IL-33 promotes ILC2 IL-9 driven mast cell responses to promote resolution of S.ratti infection.

This is a novel for identifying the role of IL-33 driving mast cell responses in this model infection. Observations are robustly tested using a range of appropriate transgenic mouse strains.

Reviewer #3: This manuscript presents a series of very well-designed experiments that demonstrate that IL-33 triggers rapid mast cell-mediated expulsion of Strongyloides ratti from the intestine, independent of the adaptive immune system, basophils, eosinophils but dependent on IL-9 receptor and innate lymphoid cells. The paper is very clearly written and significantly advances our knowledge of host parasite-relationships and intestinal immune mechanisms.

**Part II – Major Issues: Key Experiments Required for Acceptance**

Reviewer #1: 1) The authors should devise additional experiments focused on establishing the importance of endogenous IL-33 (rather than administered IL-33) on this immune response. This will help to establish the in vivo relevance of the work.

Reviewer #2: The study is great and certainly worthy of publication. However a major weakness of the study which the authors need to address is to show endogenous IL-33 production following S. ratti infection.

The study as it stands very elegantly shows that addition of an exogenous source of IL-33 drives ILC2-IL-9 dependent mast cells responses.

The authors could set the scene for the study by showing that S. ratti induces a presumably epithelial IL-33 response. This could be demonstrated by ELISA and/or histologically (RNAScope gives good resolution).

Consideration should be made to including IHC of mast cell responses throughout the manuscript to help identify changes in location as well as magnitude of mast cell responses. Is the effect of IL-9/anti-IL-33 to reduce mast cell numbers or to impair mast cell responses. cKIT staining or similar could help to address this.

Reviewer #3: None

**Part III – Minor Issues: Editorial and Data Presentation Modifications**

Reviewer #1: These were stated in Part I.

In addition, the second sentence of the introduction: "Taking advantage of the H. polygyrus-derived alarmin release inhibitor (HpARI)(29), we tested the impact of IL-33 on the early immune response to S. ratti as a model for any parasitic intestinal nematode with tissue-migrating life stages." seems overstated. I suggest removing the part of the sentence after "S. ratti".

Reviewer #2: N/A

Reviewer #3: 1. It is suggested that the terms IL-33, mast cells and Strongyloides ratti be added to the keywords.

2. Figure 1. There is large range in the results presented especially animals treated with HpARI in Panel B. It is stated that the data comes from 3 separate experiments. Did each of the 3 experiments have the same result, a wide range of responses? Did each individual experiment achieve statistical significance? The figure legend does not explain the experiments presented in panel C.

3. The legend for Figure 2 does not provide the needed information to interpret the figure.

4. The legend in Figure 4 describing the open and closed circles and squares is placed in an obscure spot.

5. It is unclear what the value of the data is on page 13 lines 1-15 and the authors should either explain what it contributes to the study or eliminate this section.

6. The authors should consider adding a figure illustrating the proposed sequence of events described on page 17 lines 6-10.

7. The paper would benefit from a concluding paragraph in which the authors provide context for the results from this study and how this new information might be applied to other infections or potentially to the development of new approaches to control infection.

PLOS authors have the option to publish the peer review history of their article (what does this mean?). If published, this will include your full peer review and any attached files.

Reviewer #1: No

Reviewer #2: No

Reviewer #3: No
---

## [Editor Report · Decision Letter 1]

2 Nov 2020

Dear PD Dr. Breloer,

We are pleased to inform you that your manuscript 'IL-33 facilitates rapid expulsion of the parasitic nematode Strongyloides ratti from the intestine via ILC2- and IL-9-driven mast cell activation' has been provisionally accepted for publication in PLOS Pathogens.

Best regards,

Meera Goh Nair

Guest Editor

PLOS Pathogens

P'ng Loke

Section Editor

PLOS Pathogens

Kasturi Haldar

Editor-in-Chief

PLOS Pathogens

orcid.org/0000-0001-5065-158X

Michael Malim

Editor-in-Chief

PLOS Pathogens

orcid.org/0000-0002-7699-2064

In this revised version, the authors have addressed the main critiques from the reviews, notably, additional assessment of endogenous IL-33 and mast cell responses. Although the authors did not specifically undertake some of the experiments requested - such as mast cell IHC and IL-33 RNAscope, they provide valid reasons why these would be very difficult to perform within a reasonable timeframe, and instead, provide alternative methods that support their conclusion. As such, this revised manuscript is improved and has addressed the reviewers' main critiques.
---

## [Editor Report · Acceptance letter]

7 Dec 2020

Dear PD Dr. Breloer,

We are delighted to inform you that your manuscript, "IL-33 facilitates rapid expulsion of the parasitic nematode Strongyloides ratti from the intestine via ILC2- and IL-9-driven mast cell activation ," has been formally accepted for publication in PLOS Pathogens.

Best regards,

Kasturi Haldar

Editor-in-Chief

PLOS Pathogens

orcid.org/0000-0001-5065-158X

Michael Malim

Editor-in-Chief

PLOS Pathogens

orcid.org/0000-0002-7699-2064